# Probiotics and Commensal Bacteria Metabolites Trigger Epigenetic Changes in the Gut and Influence Beneficial Mood Dispositions

**DOI:** 10.3390/microorganisms11051334

**Published:** 2023-05-18

**Authors:** Luis Vitetta, Matthew Bambling, Esben Strodl

**Affiliations:** 1Faculty of Medicine and Health, The University of Sydney, Sydney, NSW 2005, Australia; 2Faculty of Medicine and Health, The University of Queensland, Brisbane, QLD 4072, Australia; m.bambling@uq.edu.au; 3Faculty of Health, Queensland University of Technology, Brisbane, QLD 4058, Australia; e.strodl@qut.edu.au

**Keywords:** bio-therapeutics, probiotics, prebiotics, *Lactobacillus*, *Bifidobacterium*, major depression, gut dysbiosis, epigenetics, inflammation, immunomodulation, enteric viruses

## Abstract

The effect of the intestinal microbiome on the gut–brain axis has received considerable attention, strengthening the evidence that intestinal bacteria influence emotions and behavior. The colonic microbiome is important to health and the pattern of composition and concentration varies extensively in complexity from birth to adulthood. That is, host genetics and environmental factors are complicit in shaping the development of the intestinal microbiome to achieve immunological tolerance and metabolic homeostasis from birth. Given that the intestinal microbiome perseveres to maintain gut homeostasis throughout the life cycle, epigenetic actions may determine the effect on the gut–brain axis and the beneficial outcomes on mood. Probiotics are postulated to exhibit a range of positive health benefits including immunomodulating capabilities. *Lactobacillus* and *Bifidobacterium* are genera of bacteria found in the intestines and so far, the benefits afforded by ingesting bacteria such as these as probiotics to people with mood disorders have varied in efficacy. Most likely, the efficacy of probiotic bacteria at improving mood has a multifactorial dependency, relying namely on several factors that include the agents used, the dose, the pattern of dosing, the pharmacotherapy used, the characteristics of the host and the underlying luminal microbial environment (e.g., gut dysbiosis). Clarifying the pathways linking probiotics with improvements in mood may help identify the factors that efficacy is dependent upon. Adjunctive therapies with probiotics for mood disorders could, through DNA methylation molecular mechanisms, augment the intestinal microbial active cohort and endow its mammalian host with important and critical co-evolutionary redox signaling metabolic interactions, that are embedded in bacterial genomes, and that in turn can enhance beneficial mood dispositions.

## 1. Introduction

What has ensued from epidemiological studies is the observation of an existence of an association between psychiatric and gastrointestinal disorders. Patients diagnosed with anxiety disorders or major depressive disorders (MDD), often also have reported intestinal complaints [1]. In addition to this, patients diagnosed with intestinal disorders such as ulcerative colitis, Crohn’s disease, and irritable bowel syndrome (IBS), often present with a picture of co-morbidity with anxiety issues and/or depression [1]. Such epidemiological outcomes have introduced the concept of the gut–brain axis that informs researchers of the extensive bidirectional communication network that links the intestinal tract with central cognitive and emotional centers within the central nervous system (Figure 1).

Reports on the gut–brain axis apprise the bidirectional molecular interactions that exist between the intestines, the resident intestinal microbiome, and the enteric and central nervous systems [2]. Investigations of the intestinal microbiome have progressed our understanding of the gut–brain axis’ role in mood disorders [3]. Numerous studies on laboratory animals and humans have reported that an imbalanced intestinal microbiome (i.e., dysbiosis) is linked to mood disorders such as anxiety and depression [4,5]. Moreover, intestinal dysbiosis-associated depression can present with gut inflammatory sequalae which can significantly exacerbate excessive adverse gut mucosal barrier integrity (i.e., a dysbiotic intestinal barrier), further disrupting homeostasis [5].

The dominant phyla in a typical intestinal microbiome include the *Bacteroidetes* and Firmicutes that account for approximately 90% of the intestinal bacterial community, and other subdominant phyla that include Actinobacteria, Proteobacteria, and Verucomicrobia [6]. Research activities are accumulating evidence that the intestinal microbiota may influence brain activity and behavior via neural and humoral pathways [7]. It is further reported that gut bacteria may have translational applications in the treatment of neuropsychiatric disorders such as MDD [8]. Support for these contentions comes from several animal studies that overall conclude that intestinal bacteria could have a significant impact on the neurobiological features of mood dispositions and depression [9,10,11].

Recent reviews report that probiotic bacteria may have beneficial effects in terms of improving mood disorders [12], or that as an adjunctive treatment with pharmacotherapy may provide improved clinical psychotherapy outcomes [13]. An opinion report posited that probiotics could facilitate the epigenetic regulation of traits [14] that may be beneficial to host gut health. Epigenetic studies inform on the changes in a chromosome that result in a stable heritable phenotype without alterations in the DNA sequence [15]. Bacterial genome methylation is an area of concentrated research activities given the broad implications for genomic diversity in terms of replication fidelity, response to stress, gene expression regulation, bacteriophage resistance, and virulence [16].

The established consensus from clinical, epidemiological, and immunological evidence suggests that intestinal bacteria can extensively and profoundly influence how the gut and the brain respond and influence mental states, emotional regulation, neuromuscular function, and the regulation of the hypothalamus–pituitary–adrenal axis (HPA) [1]. The effect of probiotics on psychiatric symptoms and central nervous system function in human health remain contentious as reported in a recent meta-analysis [17]. Notwithstanding, the meta-analysis [17] concluded that developing next-generation probiotics to improve psychiatric conditions and other CNS functions is a worthwhile endeavor with early reports showing clinical efficacy. These promising signs of clinical effectiveness may be due to the epigenetic effects exerted on gut metabolites (e.g., B group vitamins, SCFAs, orotate metabolites) elaborated in the intestines by probiotic and commensal bacteria that can contribute to beneficial epigenetic changes [14].

Research continues to elucidate mechanisms of action to explain both the direct and indirect effects of intestinal bacteria on emotional and cognitive centers of the brain [7] that specifically link gut bacteria compositional changes to brain function and behavior. The intestines harbor a complex and diverse community of bacteria with a concentration gradient from the proximal to distal direction with most of the metabolic activity (i.e., host bacterial nutrient, xenobiotic and drug metabolism) restricted to the colon [18]. Reports show that the intestinal bacteria can elaborate a plethora of neurotransmitters including dopamine, gamma-amino butyrate, norepinephrine, acetylcholine, and serotonin that can affect mood [19] as well as short-chain fatty acids (SCFAs) (e.g., butyrate) and hormones such as cortisol and chemicals that modulate the immune system (e.g., quinolinic acid) [19]. For example, the Flemish Gut Flora Project observed a depletion of butyrate-producing bacteria (i.e., *Coprococcus* and *Dialister*) in individuals diagnosed with depression [19]. Whilst gut bacteria that were butyrate producers such as *Faecalibacterium* and *Coprococcus* bacteria were consistently associated with higher quality of life indicators [19].

Investigators agree that a healthy gut is related to a normal functioning central nervous system (CNS) [2,20,21,22]. It has been posited that gut dysbiosis exacerbates taxonomic changes, as observed in patients with MDD [22,23]. These changes have been associated with bacterial proinflammatory activity, impaired intestinal barrier integrity, and neurotransmitter production, with adverse effects on carbohydrates, tryptophane and glutamate metabolic pathways, and reduced levels of SCFAs [22]. An early report on gut dysbiosis from clinical sampling showed that the gut bacterial compositions of patients with major depressive disorders (MDD) versus those of healthy controls exhibited differences that were characterized by significant changes in the relative abundance of *Firmicutes*, *Actinobacteria* and *Bacteroidetes* phyla [24].

An intestinal microbiota–inflammasome hypothesis of MDD has been advanced that proposes the occurrence of pathological shifts in the composition of the intestinal microbiota (i.e., dysbiosis) that are exacerbated by stress and gut conditions that result in the upregulation of pro-inflammatory pathways [25]. Early studies support such links between gut resident bacteria and mood disorders. A study has reported an inflammasome-signaling pathway that affects anxiety and depressive mood disorders [26]. In a murine model study that induced gut inflammation with the administration of caspace-1 (an evolutionarily conserved enzyme that influences pro-inflammatory cytokines), the authors showed that when the animals were stressed and treated with an antibiotic (i.e., minocycline) the intestinal microbiota displayed alterations with increases in relative abundances of *Akkermansia* spp. and *Blautia* spp. These relative changes were compatible with the beneficial effects of attenuated inflammation and re-equilibrized gut microbiota. Moreover, the increase in *Lachnospiracea* abundance was consistent with changes in the microbiota in terms of caspace-1 deficiency [26]. Along the similar mechanistic studies, others have suggested that the emerging evidence further supports the presence of a microbiota–intestinal–inflammasome–brain axis, in which enteric bacteria modulate, via NLRP3 signaling, inflammatory pathways that, in turn, contribute and influence brain homeostasis [27].

In a recent gut microbiome-wide association study it was reported that thirteen microbial taxa, from the genera *Eggerthella*, *Subdoligranulum*, *Coprococcus*, *Sellimonas*, *Lachnoclostridium*, *Hungatella*, *Ruminococcaceae* (UCG002, UCG003 and UCG005), *Lachnospiraceae*UCG001, *Eubacterium ventriosum* and *Ruminococcus gauvreauii group*, and from the family *Ruminococcaceae* were associated with depressive symptoms [28]. The study reported that bacteria provoked a depressive phenotype through the production of serotonin and glutamate. A similar study, that investigated the association of the intestinal microbiome with depressive symptoms in a multiethnic cohort, comprising six different ethnic groups, reported depression with diversity changes in the gut bacterial cohort [29]. Selecting a wide range of confounders, the study identified genera/species belonging to the families *Christensencellaceae*, *Lachnospiraceae*, and *Ruminococcaceae* that were consistently associated with depressive symptoms across ethnicities [29]. The current thinking posits that the most consistent associations of gut bacteria with depression have been reported for genera *Eggerthella*, *Coproccocus*, *Subdoligranulum*, *Mitsuokella*, *Paraprevotella*, *Sutterella* and the family *Prevotellaceae* [30,31]. A study has also been conducted on intestinal dysbiosis that concluded that a disrupted intestinal microbiome was a key driver of MDD [32]. In this study, there were reported changes in metabolites (i.e., significantly decreased citrate and significantly increased pyruvate), particularly lipoproteins, that the authors concluded were consistent with the differential composition of the intestinal microbiota belonging to the order *Clostridiales* and the phyla *Proteobacteria*/*Pseudomonadota* and *Bacteroidetes*/*Bacteroidota*.

In a recent preliminary study, we reported the significant group differences in the relative abundance of the intestinal microbiota in MDD [33], that were observed at each taxonomical level, including across 15 genera and 18 species. This study contributes to our knowledge that there is a depressive intestinal microbial profile that is unique to the anxious distress subtype of MDD [33].

## 2. Epigenetics and Mood Disorders

Epigenetics involves signals that control DNA–protein interactions that can cause a phenotypic change in the absence of a genetic mutation [34]. Epigenetic information is genetic material that is superimposed over the existing nucleotide sequences [35]. An almost universal mechanism of epigenetic signaling is DNA methylation [35]. There are several epigenetic mechanisms that can influence genetic variations including DNA modifications (e.g., CpG methylation and demethylation), histone modifications (e.g., acetylation and deacetylation), and microRNAs functioning as translators between genes and environmental cues [36]. DNA methylation is the most stable modification that could be passed to the next generation, and DNA methyltransferases (DNMTs) are critical enzymes whose activities underlie these biochemical processes [37] (Figure 2).

Mood disorders have a highly complex and multifactorial disposition, that embraces the continuous and elaborate interplay between genetic and environmental factors. Indeed, environmental influences such as exposure to stressful tendencies can shape epigenetic patterns, and early life experiences which in certain instances can increase the risk of later-life mood disorders continue to alter the function of the genome throughout the lifespan.

In an early report on epigenetics influencing mood disorders, despite its fragmentary nature, it was proposed that DNA methylation in mood regulation was indicated by the presence of the histone deacetylase inhibitor valproate, which increases antimanic effects [38]. This notion was supported by the anti-depressive effect of *S*-adenosyl methionine, a methyl donor in DNA methylation [38]. Since then, epigenetic shifts have been associated with various mental disorders and many comprehensive reviews have been published suggesting a number of discoveries: (i) there is strong evidence that supports that all classes of psychiatric drugs modulate diverse features of the epigenome [36,39]. Schiele et al. (2020) [36] provide the first evidence that points to the transgenerational transmission of epigenetic information. Schiele et al. also report that epigenetic alterations that result from successful psychotherapy could be transferred to future generations and thus contribute to the prevention of mental disorders [36]; (ii) multi-level research data have been reported for genetic and epigenetic variation in the *OXTR* gene and social anxiety disorder, and for *CRHR1* gene variation in women with panic disorder [40]. In addition, genetic variants in the *RGS2* and *ASIC1* genes have been linked to panic disorder, while those in the ASIC1 gene are linked with treatment response in social anxiety disorder; (iii) the monoaminergic ‘risk’ genes (i.e., *SLC6A4*, *MAOA*, and *HTR1A*) were related to social anxiety disorder, generalized anxiety disorder, women with panic disorder, anxiety traits and response to psychopharmacological and psychotherapeutic interventions [40].

The future of epigenetic research includes therapeutic targets for stress-associated epigenetic changes in numerous genes correlated with depression [41]. The genes include *NRC31*, *SLCA4*, *BDNF*, *FKBP5*, *SKA2*, *OXTR*, *LINGO3*, *POU3F1* and *ITGB1* [36,41]. Furthermore, it includes genes that target epigenetic changes in glucocorticoid signaling (e.g., *NR3C1*, *FKBP5*), serotonergic signaling (e.g., *SLC6A4*), and neurotrophin (e.g., *BDNF)* genes [36,41].

Schiele and colleagues [36] focused their review on the role that epigenetic shifts have as mediators/mechanisms of psychosocial stress and psychotherapeutic interventions. They proposed that such epigenetic shifts conferred a risk of or resilience towards anxiety, affective and stress-related disorders via causing maladaptive or adaptive responses to environmental influences (Table 1). This is an improved understanding of the possible heritable role of epigenetic shifts in the complex pathogenesis of anxiety and affective and stress-related disorders and the elucidation of epigenetic mechanisms of psychotherapy may prove the benefits of the development of novel preventive measures and the benefits for the optimization of psychotherapy where there is a high rate of non-remission and treatment resistance (Table 1) [36].

With relevance to brain areas and mood disorders, it has been reported that the DNA methylation apparatus controls the dynamic regulation of methylation patterns in discrete brain regions [42,43]. Stress has been reported to be causal in the development of depression in about 60% of cases [44], where such exposure can modify DNA methylation patterns and adversely affect brain plasticity and emotion [45].

### Bacteria and Epigenetics

Reports cite epigenetic regulation and control as being increasingly recognized as a potent mechanism through which the microbiota influence host physiology and that it can occur through multiple potential mechanisms [46,47,48,49]. That is, in addition to environmental inputs (e.g., nutrition), epigenetic control includes (i) microbial biosynthesis or metabolism which influence the availability of chemical donors for DNA methylation, histone modifications or chromatin remodeling; (ii) regulation of epigenetic-modifying enzyme expression and/or activity; or (iii) activation of host cell intrinsic processes that direct epigenetic pathways such as microRNA pathways [46,50].

Many bacterial species are subject to epigenetic changes that describe DNA modifications as gene biochemical regulatory actions; that is epigenetic, signals control DNA–protein interactions and can cause phenotypic changes in the absence of a mutation. A recent review [35,49] appropriately summarized how epigenetic DNA methylations in bacteria can protect bacterial genomes, promote chromosome replication and segregation, and nucleoid organization, and control bacterial cell cycles as well as repair bacterial genome DNA and regulate transcriptional activities. Moreover, DNA methylation has been shown to control the reversible switching of gene expression [35]. This versatile action is a phenomenon that generates phenotypic cell variants [35]. Hence, the development of epigenetic bacterial lineages is important as it facilitates the adaptation of bacterial populations to severe or changing environmental conditions and modulates the interaction of prokaryotic pathogens with their eukaryotic hosts [35,49].

Research on the benefits of probiotic bacteria from major genera such as *Lactobacillus* has progressed from observational and interventional studies to the identification of the underlying molecular mechanisms that exist [14]. A regulatory mechanism of chromatin structure and gene expression is histone acetylation. Recent in vitro studies that have explored gene expression reported that probiotic bacteria such as *Lactobacillus rhamnosus* and *Lactobacillus fermentum* modulate host epigenetic signatures of intestinal epithelial cells through global histone acetylation independent of the recruitment of transcriptional activators and via *Escherichia coli* challenge [51]. In a follow-up study, this group [52] investigated temporal changes in DNA and histone modifiers (i.e., *DNMT1*, *TET2*, *p300*, *HDAC1*, *KMT2A*, *KDM5B*, *EzH2*, and *JMJD3*) and the effect on intestinal epithelial cells. They reported that in a time-dependent treatment over a 12 h period, *L. fermentum* enhanced the transcription of epigenetic modifiers (*p*  <  0.05) in intestinal epithelial cells contrary to what was observed for *Escherichia coli*, which reduced the expression of these genes with the same duration of treatment [52], whereas *L. rhamnosus* did not induce any significant changes in gene expression of the histone modifiers. The overall result was that the probiotic *L. fermentum* modulated the mRNA expression of DNA and histone modifiers contrarily to *E. coli* in a strain-specific manner [52]. However, the classical mechanisms of action that probiotics have been aligned to involve the intestinal microbiome, evoking responses that enhance the intestinal epithelial barrier and increase beneficial bacterial adhesion to the intestinal mucosa, with an attendant direct and indirect inhibition of pathogen activities [53]. Furthermore, probiotic strains have also been reported to generate a range of antimicrobial substances and to positively affect and modulate immune system function [53,54] and mood dispositions [55].

## 3. Molecules That Influence the Gut–Brain Link

It is known that the intestinal microbiota affects many physiological processes, such as cell proliferation, epithelial barrier function, and immune responses, and these processes have a direct link to stress and mood dispositions [56]. The role of probiotics in the management of mood appears complex, with the evidence being stronger in depressed populations than in healthy populations [57]. However, this complexity might be explained by the molecules displaying epigenetic influence that have been posited to play important roles in mediating the health-promoting attributes of commensal and probiotic bacteria [14].

### 3.1. Reactive Oxygen Species

Reactive oxygen species (ROS) such as superoxide and hydrogen peroxide [58] are the active intermediates and regulators of major epigenetic processes such as DNA methylation and histone acetylation reactions [59]. Early reports show that specific taxa of intestinal bacteria can induce the rapid and transient enzymatic production of reactive oxygen species (ROS) within enterocytes [60]. This report showed that redox signaling can be triggered by microbially generated ROS. Mechanistical formation was reported to be a response to microbial signals activated via formyl peptide receptors and the gut epithelial NADPH oxidase1 (Nox1) [60]. There are different strains of commensal gut bacteria that can elicit markedly different levels of ROS from cells that they adhere to. Therefore, the idea has been presented that there is favorable redox potential that the gut microbiota (i.e., the tendency and capacity of the microbiota to gain electrons) and the host have which influences the homeostasis of the intestinal barrier [61]. Studies that administered antibiotics demonstrated how altering the diversity of the gut microbiome through antibiotic-induced changes can disrupt redox dynamics in the intestines [61]. It has also been posited that the redox potential in the intestines can also be modulated by the brain and the CNS via the vagal cholinergic anti-inflammatory pathway [62,63]. For example, *Lactobacillus* species from the gut have been reported to specifically generate potent inducers of ROS generation in cultured cells as well as in vivo. Moreover, researchers have reported that numerous bacteria exhibit some ability to alter the intracellular oxido-reductase environment [64].

Both DNA methylation and histone acetylation are nucleophilic processes and therefore ROS signaling through typical free-radical processes can take part in epigenetic processes via the reactions of nucleophilic substitution, and these reactions represent the first explanation of their role in epigenetic processes [65,66].

Yan et al., (2007) reported that there are soluble factors that are produced by different strains of *Lactobacillus* that can have beneficial effects in in vivo inflammatory models [67], thereby presenting data that expand our understanding of the intestinal microbiome’s activity. These results indicate that there are ROS-stimulating bacteria that possess effective specific membrane components and or secreted factors that activate cellular ROS production to maintain homeostasis [68]. Furthermore, it has been reported that enteric commensal bacteria produced the rapid generation of ROS thereby negotiating acceptance from the intestinal epithelia [69].

### 3.2. B Group Vitamins

The metabolic pathways of B vitamins (i.e., vitamin B_6_ and B_9_ and B_12_) have continuously been implicated in DNA methylations [59], where deficits of these vitamins have been thought to contribute to cognitive decline through increased homocysteine levels with subsequent adverse effects on oxido-reductase redox signaling [70]. In a recent longitudinal study with 2533 participants [71], it was concluded that higher levels of dietary folate at the baseline predicted a better cognitive reserve. Alternatively, decreased serum levels of B vitamins were posited to contribute to cognitive impairment, and this was in turn reported to affect the level of methylation of specific redox-related genes [71]. In a recent systematic review, it was concluded that in at-risk populations with poor mood dispositions, vitamin B group supplementations may provide a significant health benefit [72].

Furthermore, it has been reported that *Streptococcus thermophiles*, *Lactobacillus* and *Bifidobacterium* genera synthesize folic acid to increase DNA methylation and mRNA N6-methyladenosine in the gut, to thereby maintain normal intestinal homeostasis [73,74]. These data support the position that epigenetics presents a new frontier for the beneficial health effects that probiotics portray beyond the data reported by interventional studies [14]. The underlying molecular mechanisms associated with the health-promoting effects of probiotics are linked to effector molecules produced by probiotics that influence specific genes and even individual nucleotides [75].

### 3.3. Short-Chain Fatty Acids (SCFAs)

Although the underlying mechanisms remain unclear, SCFAs, the chief colon metabolites produced via commensal bacterial fermentation of dietary fibers and resistant starch (Table 2), are postulated to have key roles in mediating the relationship between the intestines and DNA methylation [76,77]. For example, the SCFA butyrate has been reported to improve adverse mood dispositions [78].

The genome of some species of the *Bifidobacterium* genera from the human gut and those also of animal origin demonstrate a high presence of genes involved in the metabolism of complex oligosaccharides [79]. Other resident bacteria of the colon are also able to degrade inulin-type fructans, as is the case for *Lactobacilli*, *Bacteroides*, certain enterobacteria, and butyrate producers (Table 2). Bacterial cross-feeding mechanisms in the colon form the basis of overall butyrate production, a functional characteristic of several colon bacteria. In addition, the specificity of polysaccharide use by the colon microbiota may determine diet-induced alterations in the microbiota and consequent metabolic effects [80]. Certainly, supplementation with undigested polysaccharides of plant origin is important for the enrichment of the intestinal microbiota with *Lactobacillus* and *Bifidobacterium* species, which ferment these compounds into SCFAs [81,82]. Early research has reported that SCFAs, such as butyrate, can have epigenetic effects on the gut [83]. Although the mechanism of action of butyrate is complex, numerous actions involve an epigenetic regulation of gene expression through the inhibition of histone deacetylase [83].

**Table 2 microorganisms-11-01334-t002:** Probiotic and commensal/intestinal bacteria producing SCFAs (adapted from multiple published sources [81,84,85,86,87,88]).

Types of Bacteria	SCFAs Produced
Probiotics	
*Bifidobaterium* genus	
*Bifidobacterium* spp.	Acetate|Lactate
*Bifidobacterium longum*, *Bifidobacterium bifidum*	Acetate|Lactate|Propionate
*Lactobacillus* genus	
*Lactobacillus rhamnosus* GG (LGG), *Lactobacillus gasseri*	Lactate|Propionate
*Lactobacillus acidophilus*	Acetate|Butyrate|Lactate|Propionate
Commensal|Intestinal Bacteria	
*Dalister succinatiphilus* *Eubacterium* spp. (e.g., *E. halli*) *Megasphaera elsdenii* *Phascolarctobacterium succinatutens* *Roseburia* spp. *Salmonella* spp. *Veillonella* spp.	Propionate
*Coprococcus* spp. (e.g., *Coprococcus catus*) *Roseburia inulinivorans*	Butyrate|Propionate
*Clostridium* spp. *Ruminococcus* spp.	Acetate|Butyrate|Propionate
*Anaerostipes* spp. *Coprococcus comes*, *Coprococcus eutactus*, *Clostridium symbiosum* *Eubacterium rectale*, *Eubacterium hallii*, *Faecalibacterium* spp. (e.g., *Faecalibacterium prausnitzii*) *Roseburia* spp. (e.g., *Roseburia intestinalis*)	Butyrate

A mechanistic review reported that butyrate [89] and beta-hydroxy-butyrate (b-OHB) [90] are inhibitors of the catalytic activity of Zn^2+^-dependent histone deacetylases (HDACs). Such inhibition was robustly shown to elicit anti-inflammatory effects in cell culture and rodent model studies. Specifically, SCFA inhibition of HDACs was beneficial in improving neurocognitive and mood disorders. The class I HDAC inhibitor MS-275 prevented depression-like behaviors in mice when subjected to a social stress model [90].

There has been a stream of continuous research that has focused on the role fatty acids have in the neurometabolic pathophysiology of psychiatric disorders [91]. Furthermore, brain phospholipid metabolism and membrane fluidity have been posited to be involved in mood disorders [92]. In a murine model using the forced swim test reported that dietary supplementation with omega-3 fatty acids reduced immobility when given for 30 days [92]. This study also investigated and postulated the antidepressant effect of uridine. Moreover, when both agents were administered together, less of each agent was required for an effective antidepressant outcome.

### 3.4. Bile Salts, Mood Dispositions and Intestinal Bacteria

We previously reported that the intestinal microbiome can significantly affect and dysregulate the biochemistry of bile salts [93]. A systematic review of gut microbiome variations involved an investigation on patients diagnosed with MDD [94]. From the 17 included studies in the systematic review, 4 found that there was reduced alpha diversity in studies of MDD. Further, intestinal microbiota compositions were clustered separately according to the β diversity between patients and controls in twelve other studies. When comparing patients with MDD versus controls, it was concluded that there was an increase in the relative abundance of the bacterial genera *Eggerthella*, *Atopobium*, and *Bifidobacterium*, and that there was a decrease in the relative abundance of *Faecalibacterium* in MDD patients compared to the controls [94].

A recent study with a chronic unpredictable mild-stress-induced mouse model showed that increased secondary bile acid levels in the feces positively correlated with *Ruminococcaceae*, *Ruminococcus*, and *Clostridia* abundances [95]. The authors concluded that an increased abundance of bacterial species from the family *Ruminococcaceae* responded to chronic stress with an increased level of biosynthesis of deoxycholic acid, an unconjugated secondary bile acid in the intestines.

Moreover, a recent study on patients diagnosed with MDD compared to healthy controls reported that there exists a disturbance in the intestinal microbiome of patients diagnosed with MDD [96]. It was also reported that gut dysbiosis correlated with a disturbance in bile salt metabolism [96]. Specifically, bile salt analysis showed that the amount of 23-nordeoxycholic acid in patients with MDD was significantly higher than that in healthy controls, whereas the amounts of 9 taurolithocholic acid, glycolithocholic acid, and lithocholic acid 3-sulfate were significantly lower. Moreover, the intestinal microbiome showed positive associations between *Turicibacteraceae*, *Turicibacterales*, and *Turicibacter* with taurolithocholic acid, glycolithocholic acid and glycodeoxycholic acid [96].

### 3.5. Choline, Trimethylamine-N-Oxide and Bacteria

Dietary choline is an important nutrient for the production of acetylcholine, a neurotransmitter that plays an important role in regulating memory, mood, and intelligence [97] as well as in the synthesis of methionine, a methyl donor of s-adenosyl methionine (SAMe) [98]. Hence, choline is a metabolite that has been reported to be a methyl donor that influences epigenetic regulation [98].

There appears to be a paradoxical view in terms of the levels of choline relative to mood that are found in different body compartments. In a review analysis of clinical studies, it was reported that high levels of choline concentration in the frontal lobe were associated with depression both in those who responded to treatment and those who did not, following treatment with psychiatric medications, repetitive transcranial magnetic stimulation, or electroconvulsive therapy [99]. In contrast, in a large observational study, it was concluded that the lowest choline quintile level in blood was significantly associated with high anxiety levels but not depression [100]. Furthermore, Romano and colleagues (2017) reported that a gut bacteria-induced reduction in choline reduced methyl-donor availability and influenced global DNA methylation patterns with alterations in behavior.

SAMe is involved in one-carbon metabolism and epigenetic modifications of DNA [101]. The administration of SAMe has given positive data in the management of MDD when administered as a monotherapy or as an adjunct to antidepressant pharmacotherapy [102]. In a recent review, the therapeutic roles of SAMe and probiotics in depression was highlighted [103]. SAMe and probiotics may have synergistic effects in terms of positively influencing mood dispositions.

Intestinal bacterial choline metabolism generates trimethylamine oxide (TMAO) and regulates epigenetic mechanisms [48,104]. A recent study reported that elevated TMAO levels immediately after an AMI could reflect severe stress in PTSD-vulnerable patients [105], making it a plausible biological correlate for severe stress that is associated with vulnerability to PTSD.

### 3.6. Orotate, Uridine and Bacteria

Orotate is an organic compound present in dairy products as well as in the milk of ruminants whereby it has critical value in basal processes of the organism. It is primarily an intermediate for pyrimidine metabolism, a precursor of uridine-monophosphate, with important roles in DNA and RNA synthesis and anti-inflammatory activity through uridine-monophosphate formation [106]. Our group conducted two pilot studies that investigated the administration of magnesium orotate in combination with SSRIs and of probiotics and SSRIs, respectively [107,108], to patients diagnosed with major depression not responsive to the SSRIs.

The first pilot study [107] investigated the plausible beneficial effects of S-adenosylmethionine (SAMe) and magnesium orotate in patients with major depression. The study was conducted with patients during a 15-week period beginning with SAMe administration. At the end of week 15, non-responsive patients with depression were subjected to an 8-week-long administration of a combined intervention of SSRIs and magnesium orotate. The results showed that there was good compliance with no participant drop-outs and a lack of adverse effects. We posited at the time that the intestinal microbiome was complicit in both the suboptimal response to psychotropic therapy and in improving the SAMe and magnesium orotate response via metabolism and absorption functions.

The second pilot study [108] recruited patients with treatment-resistant symptoms of depression. The study recruited 12 patients who were administered, over an 8-week period, a combination of magnesium orotate and probiotics as adjuncts to SSRIs. At the end of the 8-week treatment period, all participants had improved depression and anxiety scores, with self-assessment scores revealing increased energy and higher levels of well-being. These results suggest the bidirectional synergic mechanism of the combination of a SSRI, magnesium orotate and probiotics on the gut–brain axis. At the 16-week follow-up after the cessation of the administration of magnesium orotate and probiotics, there was an observed relapse of depression symptoms while the SSRI medications were still being administered [108].

Early in vitro studies demonstrated how in yeast and in extracts from livers of several species, there was an observed conversion of orotate into pyrimidine nucleotides [109]. Previous studies showed that a *Lactobacillus* species (*Lactobacillus bulgaricus* 09) for which orotate is a specific growth factor was found to have an enzyme converting orotate to uridine-5-phosphate [110], while an additional *Lactobacillus* species (*Lactobacillus arabinosus*) required uracil specifically at a certain phase of its growth cycle [110,111]. It was concluded that these bacterial species which responded nutritionally to either pyrimidine were found to have both the orotate- and uracil-utilizing enzymes [109]. Given the role of uridine derivates such as that of uridine diphosphate (UDP)-glucose 6-dehydrogenase (UGDH) in DNA methylation [112], the findings from our two pilot studies point to specific probiotic bacteria and other intestinal commensals utilizing orotate to elaborate uridine metabolites that link uridine’s role in DNA methylation.

## 4. Discussion

This narrative review has focused on molecules and metabolites that bacteria elaborate. These bacteria, in doing so, may provide the significant epigenetic regulation that is important in preventing or progressing mood disorders with effects on the gut–brain axis. Epigenetic regulation is a gene-silencing event for gene expression that in turn influences protein levels [113,114]. For example, SCFAs, especially butyrate, are well-known histone deacetylase inhibitors that have been shown to improve mood [78]. Similarly, other metabolites such as folates administered as adjuvants to SSRI pharmacotherapy improved depression scale scores, patient response, and remission rates [115]. Moreover, contrasting effects have been reported such as the microbially induced reduction in elevated levels of choline (i.e., improving mood) [99] and the production of trimethylamine-*N*-oxide (i.e., exacerbating mood disorders) by microbially influenced metabolites that can also regulate epigenetic mechanisms [98].

Our group previously postulated that the intestinal microbiota could affect neuronal mitochondrial function through short-chain fatty acids such as butyrate [116]. Anderson and Maes simultaneously reported that increasing intestinal microbiome-derived SCFAs, such as butyrate, can induce beneficial effects on mitochondrial function [117]. Such effects would allow enhanced immunologic and metabolic homeostasis control across local (i.e., the gut) and other body sites including the brain [117]. Mechanistically, butyrate’s influence on mitochondrial function includes an increase in mitochondria-located sirtuin-3. This deacetylate effect then disinhibits the pyruvate dehydrogenase complex, which would lead to an increased conversion of pyruvate into acetyl-CoA (i.e., TCA cycle), consequently increasing ATP production.

There is biological plausibility that the effects of SCFAs such as butyrate on intestinal epithelial cells can upregulate the mitochondrial melatonergic pathway [118]. Given the importance of the intestines with other end organs via numerous gut axes, the gut microbiome may exert influential effects on diverse diseases and conditions through its impact on mitochondrial function [119]. Such a notion may provide the foundation that links depression with many other medical conditions.

In a study using the murine model of depression, it was demonstrated that the antidepressant-like effects of uridine and omega-3-fatty acid showed the maximum antidepression-like effects when administered together [92]. We previously postulated that the organic molecule orotate as an intermediate metabolite in uridine metabolism may have direct links to the brain via the systemic circulation through the elaboration of uridine-5-phosphate in the intestines by the gut microbiome, producing an antidepressant effect. The relevance of this is supported by an early report that showed that uridine is an important and major form of pyrimidine nucleoside that is taken up by the brain [120]. In the brain, pyrimidine nucleoside is used to elaborate nucleic acids and for the synthesis of membrane constituents such as neuro-receptors. Plasma membrane receptors of seven transmembrane domains have been identified that recognize uridine triphosphate (UTP), uridine diphosphate (UDP), and UDP-sugar conjugates [120]. Research has reported that the effects of uridine on brain structures and functions appear to be mediated by its effects of promoting neuronal membrane formation and through interactions with specific uridine-nucleotide receptors (i.e., brain P2Y2 receptors) that control neuronal differentiation [120,121]. Additionally, UTP can be biochemically converted to cytidine triphosphate an important key intermediate in the generation of phosphatidylcholine for the synthesis of neural membranes [120].

Recent systematic reviews consistently found support for the link between the DNA hypermethylation of brain-derived neurotrophic factor (BDNF) and depression [122,123,124]. It is of interest to note that several metabolites posited in this paper, that are important in explaining the link between the gut microbiome, DNA methylation and depression, have already been identified as relevant to the DNA methylation of BDNF. For example, NOX1-derived ROS have been shown to oxidize the N-methyl-D-aspartate (NMDA) receptor affecting the DNA methylation of the BDNF gene [125]. In addition, sodium butyrate has been implicated in the processes involved in the DNA methylation of BDNF [126]. Similarly, concentrations of vitamins B_6_, B_9_ and B_12_ have been linked with BDNF expression [127]. Such research lends support for the proposed pathways proposed in this paper.

Given the emerging evidence base supporting the causal link between the gut microbiome and psychiatric disorders such as depression [30,31], there is a need to articulate the precise mechanisms contributing to the gut–brain axis pathways. DNA methylation is a strong contender of being considered an important mediator between the dysbiosis of intestinal microbiota and the experience of psychiatric disorders. There is emerging evidence to support the proposed mechanism that alterations in the concentrations of certain gut bacteria may result in changes in cellular concentrations of key metabolites involved in DNA methylation cycles. We propose that ROS, B-group vitamins, SCFAs, choline and orotate are likely critical metabolites involved in the gut–brain axis pathways linking the gut microbiome with psychiatric disorders. Indeed, similar pathways have already been identified in other health conditions. For example, in obesity-prone individuals, the gut microbiota results in decreased total SCFAs but enriched propionate, which in turn induces specific DNA hypermethylation predisposing obesity-prone individuals to diabetes [105]. Such examples from other health conditions support our call for further investigation of the pathways proposed in this paper. Greater understanding of these pathways will guide the precise selection of probiotics and prebiotics to enhance the effectiveness of probiotic interventions for clinical depression and a range of mental health conditions.

It has been suggested that probiotics can influence serotonin levels in plasma [128]. This is important given that a low number of serotonin receptors has been linked with the development of depression [129]. This group documented that the people diagnosed with depression essentially had fewer serotonin receptors throughout the brain and significantly fewer receptors in key areas such as the hippocampus, a region of the brain that acts as a gateway between memory and mood, and numerous other processes [129,130]. The same group investigated serotonin 5-HT(2A) receptors and what role these receptors have in the regulation of brain biochemical functions that are disturbed in patients with major depressive disorder [131]. The authors concluded that altered serotoninergic function in the hippocampus which was the likely outcome involved in the disturbances of mood regulation in major depressive disorder [131]. In a study using a murine model of depression, it was shown that the administration of probiotics increased the plasma levels of tryptophan, the precursor molecule for serotonin [132]. Moreover, together with another research group [133], it was reported that reduced levels of serotonin’s main metabolite, 5-hydroxyindoleacetic acid, had effects that were similar to those of the antidepressant citalopram.

## 5. Conclusions

The gut–brain axis manifests a potential link between the intestines and the central nervous system by way of a bidirectional biochemical network of communication. Emotional and cognitive centers of the brain are thus linked to peripheral intestinal functions. Consequently, in vivo research has documented and demonstrated that disorder in the intestinal microbial system’s structure is positively correlated with mood dispositions such as depression. Commensal intestinal bacteria and exogenous administered bacteria (e.g., probiotics from the *Lactobacillus* and *Bifidobacterium* genera) have been documented to influence mood dispositions. Clinical studies involving the administration of probiotics and other molecules have shown that this leads to secondary complex and overlapping mechanisms that produce metabolites (e.g., SCFAs and uridine metabolites) with epigenetic activity that influence brain activities and mood dispositions.

Narrative reviews due to their very nature have limitations. Narrative reviews report and summarize research from included studies, but do not provide direct comparisons between clinical studies. Further, as such, narrative reviews do not conduct pooled analysis. Narrative reviews however do provide a platform for significant plausible research ideas and evidence that can be developed with enhanced clinical trial designs for future robust studies that will promote clinical understanding and the development of treatments.

## Figures and Tables

**Figure 1 microorganisms-11-01334-f001:**
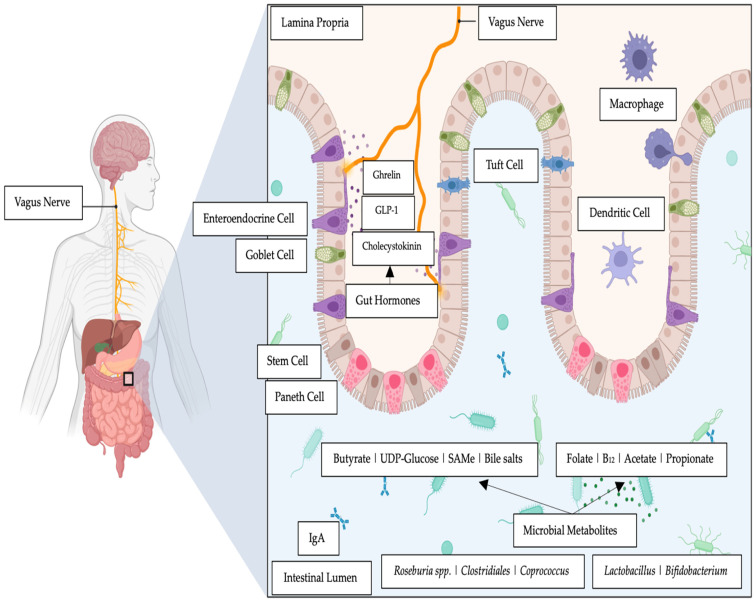
The gut–brain axis and examples of intestinal metabolites elaborated by relevant microbial species and gut hormones.

**Figure 2 microorganisms-11-01334-f002:**
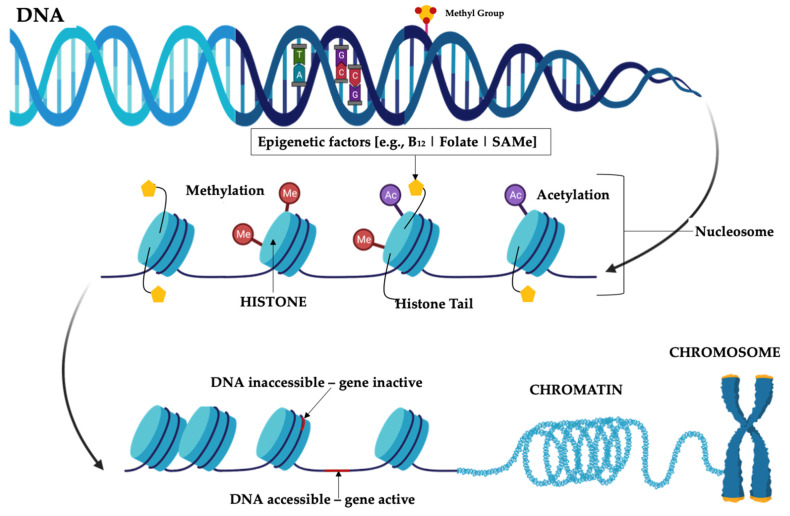
Diagrammatic representation of epigenetic mechanisms. Note: DNA methylation region is where a methyl group can tag DNA and can then activate or repress genes. Histone modification region is the site where epigenetic factors bind to histone tails which alters the extent to which DNA is wrapped around histones and the availability of genes in the DNA to be activated.

**Table 1 microorganisms-11-01334-t001:** Summary of primary findings from psychotherapy epigenetic studies in adults (adapted and modified from Schiele et al., 2020) [36].

Type of Diagnosis *	Type of Treatment *	Gene *	Primary Outcome
PTSD	PE	*NR3C1*	↑ *NR3C1* methylation in responders|post-psychotherapy
	PE	*FKBP5*	↓ *FKBP5* methylation in responders|at follow-up ↑ *FKBP5* methylation in non-responders|at follow-up
PTSD	MBSR	*FKBP5*	↓ *FKBP5* methylation in responders|post-MBSR ↑ *FKBP5* methylation in non-responders|post-MBSR
Mixed AD	CBT	*FKBP5*	↓ *FKBP5* methylation associated with ↑reduction in symptom severity|follow-up in FKBP5 rs1360780 T-allele carriers
Mixed AD	CBT	*SLC6A4*	↑ *SLC6A4* methylation in responders|at follow-up ↓ *SLC6A4* methylation in non-responders|post-CBD
PD	CBT	*MAOA*	↑ *MAOA* methylation in responders|post-CBT
SpP	CBT	*MAOA*	↑ *MAOA* methylation in responders|post-CBT
BPD	DBT	*BDNF*	↓ *BDNF* methylation in responders|post-DBT
BPD	DBT	*BDNF*	↓ *BDNF* methylation in responders
BPD	DBT	*MCF2*	↑ *MCF2* baseline methylation in responders
MDD	CBT	*SLC2A1*	↓ *SLC2A1* methylation in remitters
PD	CBT	EWAS ^⊥^	↑ *IL1R1* methylation post-CBT
PTSD	tf-CBT [partial with EDMR]		Changes in 12 DMRs ↑ *ZFP57* methylation in responders|at follow-up

* PTSD = Post-traumatic stress disorder; MBSR = mindfulness-based stress reduction; AD = anxiety disorder; PD = panic disorder; PE = prolonged exposure therapy; SpP = specific phobia; BPD = borderline personality disorder; DBT = dialectic behavioral therapy; tfCBT = trauma-focused psychotherapeutic intervention; MDD = major depression disorder; NR3C1 = nuclear receptor subfamily 3 group C member 1; FKBP5 = FKBP prolyl isomerase 5; SLC6A4 = serotonin transporter and solute carrier family 6 member 4; MAOA = monoamine oxidase A; BDNF = brain derived neurotrophic factor; MCF2 = MCF2 cell line-derived transforming sequence gene; SLC2A1 = solute carrier family 2 member 1 gene; EDMR = differentially methylated region; ZFP57 = zinc finger protein 57 homolog. ⊥ EWAS = epigenome-wide association studies (EWAS) that investigate the association between a phenotype and epigenetic variants, most commonly DNA methylation [42].

## Data Availability

Not applicable.

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
