# Peer review of "Probiotics and Commensal Bacteria Metabolites Trigger Epigenetic Changes in the Gut and Influence Beneficial Mood Dispositions"

_microorganisms, 2023, doi:10.3390/microorganisms11051334_

Round 1
Reviewer 1 Report
See attached file

Author Response
See attached file please.

Reviewer 2 Report
The manuscript by Luis Vitetta et al. summarized the probiotics and commensal bacteria metabolites that could trigger epigenetic changes in the gut and influence the beneficial mood dispositions. This review article is of interest to the readers and I have the following suggestions and comments:
1, in Figure 1, the authors should specifiy which bacterial metabolites were included in the gut-brain axis. Besides, the bacteria that could affect the gut-brain axis should also be specified. The authors must further revise.
2, I suggest the authors to add a new figure to show all kinds of epigenetic modifications. This could help the readers to understand the main idea of the whole manuscript.
3, bile acids could also affect the epigenetics of the gut-brain axis. This must be added and discussed.
4, future development and current limitations of the studies must be discussed.
Author Response
See attached please

Round 2
Reviewer 2 Report
The authors have revised the manuscript accordingly. I therefore suggest to accept it.